# Element Abundances in Impulsive Solar Energetic-Particle Events

Donald V. Reames 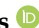

Institute for Physical Science and Technology, University of Maryland, College Park, MD 20742, USA; dvreames@gmail.com

**Abstract:** Impulsive solar energetic-particle (SEP) events were first distinguished as the streaming electrons that produce type III radio bursts as distinct from shock-induced type II bursts. They were then observed as the surprisingly enhanced $^3$He-rich SEP events, which were also found to have element enhancements rising smoothly with the mass-to-charge ratio $A/Q$ through the elements, even up to Pb. These impulsive SEPs have been found to originate during magnetic reconnection in solar jets where open magnetic field lines allow energetic particles to escape. In contrast, impulsive solar flares are produced when similar reconnection involves closed field lines where energetic ions are trapped on closed loops and dissipate their energy as X-rays, $\gamma$-rays, and heat. Abundance enhancements that are power laws in $A/Q$ can be used to determine $Q$ values and hence the coronal source temperature in the events. Results show no evidence of heating, implying reconnection and ion acceleration occur early, rapidly, and at low density. Proton and He excesses that contribute their own power law may identify events with reacceleration of SEPs by shock waves driven by accompanying fast, narrow coronal mass ejections (CMEs) in many of the stronger jets.

**Keywords:** solar energetic particles; solar system abundances; solar jets; solar flares; shock waves; coronal mass ejections

## 1. Introduction

The idea that there must be two different sources of solar energetic particles (SEPs) was expressed very early in a 1963 review of solar radio observations by Wild, Smerd, and Weiss [1]. The rapid frequency decrease in radio type III bursts is produced when fast (10–100 keV) electrons excite density-dependent plasma frequencies as they stream out from sources in the solar corona, while frequencies in the slower type II bursts evolve at the speed of coronal shock waves known to be capable of accelerating high-energy protons and other ions. Once it became possible to measure electrons in space, Lin [2] found prompt bursts of streaming ~40 keV electrons associated with ~40 keV solar X-ray events, accompanying type III radio bursts, while relativistic electrons were only seen during large energetic-proton events, with associated type II and IV radio bursts. Lin [2] believed that type III bursts could involve "pure" solar electron events, i.e., without ions.

Meanwhile, the observations of protons had begun at the highest energies, GeV protons, with nuclear cascades through the Earth's atmosphere that produced ground-level enhancements (GLEs) of residual muons above the similar background produced by galactic cosmic rays (GCRs) [3]. Abundances of dominant elements were first observed in 1961 using nuclear emulsions flown on sounding rockets by Fichtel and Guss [4] and were subsequently extended up to the element Fe [5]. Relative abundances of ions in large SEP events would become a reference for studying the physics of SEP acceleration when Meyer [6] linked the average abundances of elements in SEPs to abundances of the solar corona, which differ from those in the photosphere as a function of their first ionization potential (FIP). As the elements begin their journey from the photosphere to the corona, high-FIP (>10 eV) elements are initially neutral atoms, while low-FIP elements are ionized

and subject to electromagnetic forces, which enhance them by a factor of about three, before all elements become highly ionized in the hot corona. Abundances in individual SEP events, relative to the average coronal abundance, differ as a power-law function of the mass-to-charge ratio $A/Q$ of the ions [7] largely because of magnetic-rigidity-dependent scattering after acceleration. Recent measurements confirm the average coronal abundances and the "FIP-effect" [8–10] for SEP events, which, incidentally, differ from those of the solar wind [11–14].

After many years of controversy [10], these large SEP events that provide a basis for the coronal abundances of the elements were shown by Kahler et al. [15] to have a 96% correlation with shock waves driven out from the Sun by fast, wide coronal mass ejections (CMEs). This "large-scale shock acceleration" [16], foreseen by Wild, Smerd, and Weiss [1] so long before, was obscured for many years by the "solar flare myth" as described by Gosling [17,18]. Observations by the STEREO spacecraft now show that these shock waves and the related "gradual" SEP events can span nearly 360° in solar longitude [19]. These huge shock waves easily cross magnetic field lines, accelerating and transporting SEPs where the SEPs alone cannot go. These shock waves tend to sweep up a sample of average coronal abundances, after which differences in transport between elements, such as Fe and O, which cause enhancements of Fe/O in some regions, will cause compensating Fe/O depressions in others that tend to average out. Review articles describe impulsive SEP events [20–22], gradual SEP events [23–28], and compare both [10,29–36].

## 2. $^3$He-Rich Events

In the early days of SEP measurements in space, nearly every scientist involved had previous experience with GCRs where interstellar nuclear fragmentation of $^4$He produces significant abundances of $^2$H and $^3$He and fragmentation of C, N, and O produces Li, Be, and B. Otherwise these secondary ions would have very low abundances. Thus, it was no surprise when $^3$He/$^4$He of ~2% was first detected in SEPs [37]; solar $^3$He/$^4$He $\approx 5 \times 10^{-4}$. Surely this could come from fragmentation in the corona? But a subsequent event, measured with $^3$He/$^4$He = 1.52 $\pm$ 0.10 and $^3$He/$^2$H > 300 [38], certainly could not. Fragmentation was completely ruled out when SEP events were found to have Be/O and B/O < 2 × $10^{-4}$ [39,40]. This was not fragmentation at all; it involved a new acceleration mechanism in these "impulsive" events.

These $^3$He-rich events also had element abundances soon found to increase with elements up through Fe [41,42]. In fact, when it became possible to measure groups of even-heavier elements up to Au and Pb, enhancements relative to coronal abundances were found to continue their increase on average as the ~3.6 power of $A/Q$ [43–46], if using $Q$ values based upon an assumed source coronal temperature of ~3 MK.

### 2.1. Properties and Associations

The tie between electrons and ions in impulsive SEP events came with the unexpected association of $^3$He-rich events with non-relativistic electron events [47] and with type III radio bursts that could be tracked from the Sun [48]. Those allegedly "pure" electron events turned out to be $^3$He-rich events despite greatly enhanced electron/proton ratios. Later observations of the association of radio type III bursts and electrons with $^3$He-rich events by Nitta et al. [49] are shown in Figure 1. These authors traced the Parker spiral for the interplanetary magnetic field with the potential field source surface (PFSS) model to determine the field configuration shown in Figure 1e for each of three $^3$He-rich events. The figure shows the ion intensities, X-ray intensities, radio type III bursts, and electron intensities at various energies. The latter shows velocity dispersion (i.e., fast electrons arrive before slower ones) with onset times ordered as $L/v$, where $L$ is the path length from the Sun and $v$ is the electron velocity providing timing and association of electrons and $^3$He [47,48]. Frequencies of radio type III decrease as the square root of the local density, excited when 10–100 keV electrons from the event stream out from the Sun [47,48]. Type III

bursts can be used to track the trajectory of the electron distribution accelerated with the $^3$He ions [48]. $^3$He-rich events and spectra have been reviewed by Mason [20].

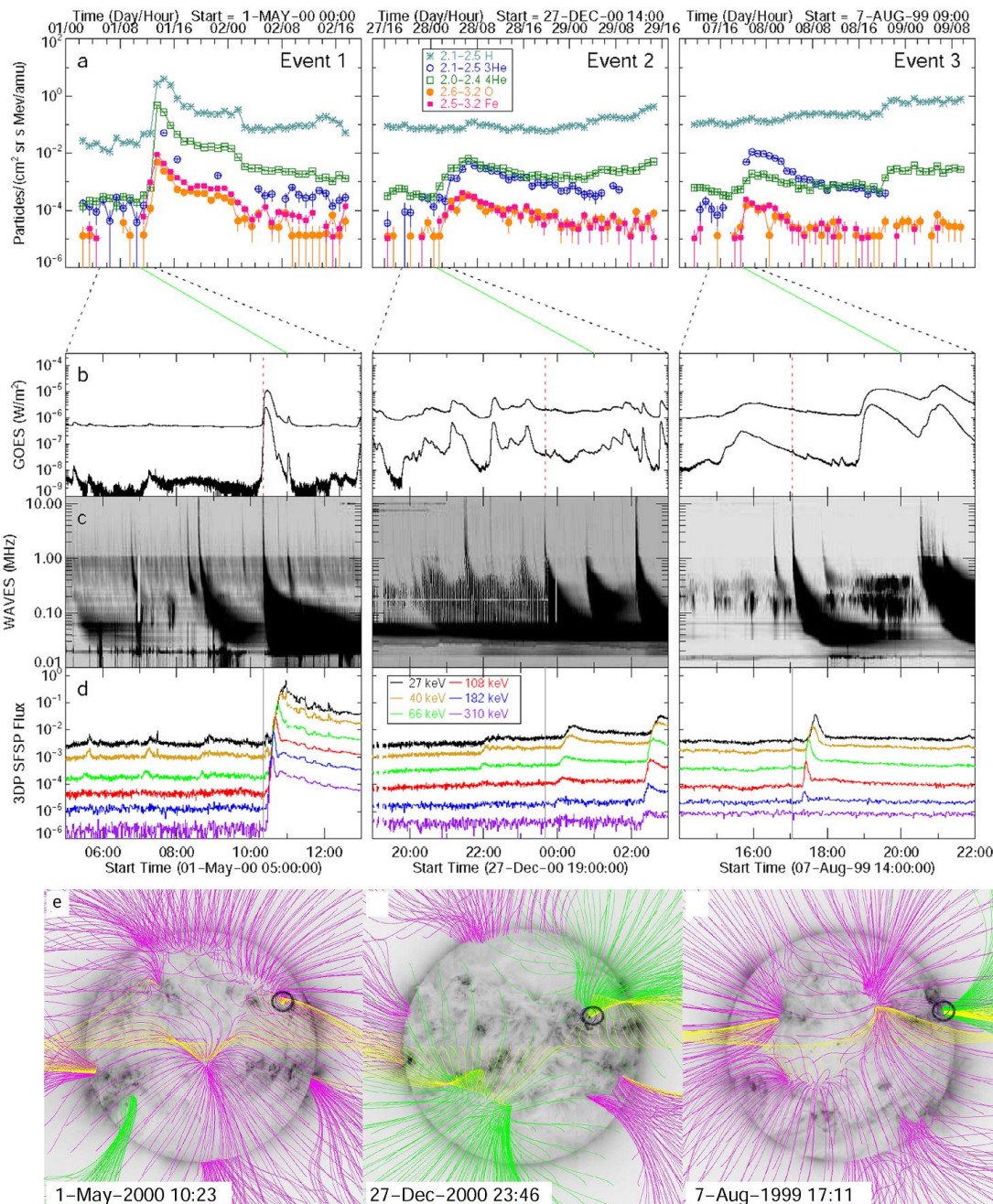

**Figure 1.** For three $^3$He-rich events, time histories are shown for (**a**) intensities from *Wind* EPACT of the listed ions in the MeV amu$^{-1}$ intervals, (**b**) GOES X-ray fluxes at (1–8 Å and 0.5–4 Å), (**c**) *Wind* WAVES radio spectra, and (**d**) *Wind* 3DP electron intensities at the listed energies. The dashed red lines mark the probable event onset times. Panels (**e**) show the PFSS model field lines. **Pink** and **green** lines show negative and positive footpoint polarities, respectively. **Yellow** marks "open" field lines that reach the source surface at 2.5 R$_S$, and black circles mark the event sources [49].

As intensities of impulsive SEP events grow larger, $^3$He/$^4$He tends to decrease and the $^3$He intensity saturates. This occurs when all the available $^3$He in the source volume begins to be exhausted, as suggested in an early review [32] and subsequently shown conclusively by Ho et al. [50]. This is also considered as a basis for the unusual energy spectrum of

[3]He [20]; waves that resonate with gyrofrequencies of the dominant H and [4]He ions tend to produce power-law energy spectra while the waves become damped, but [3]He is too rare to significantly damp waves that resonate with its isolated gyrofrequency, so it is depleted but continues to absorb energy as modeled by Liu, Petrosian, and Mason [51–53].

Most [3]He-rich events are indeed single events with clear evidence of velocity dispersion in both electrons and the ions [47], but the second two events in Figure 1c show no GOES Xray increase, suggesting minimal electron trapping (e.g., Event 3 in Figure 1, which shows no GOES X-ray increases, only SEPs and type III burst). However, some "events" encountered involve repetitive injections from a single active-region source [54–57] or are even a blur of smaller events that cannot be resolved [20,58–61]. These events commonly form "pools" or streams of suprathermal ions that can act as a pre-accelerated "seed" population with the abundance characteristics of impulsive SEPs that may become available for reacceleration by large shock waves in some gradual SEP events. This can blur the abundance distinction between impulsive and gradual SEP events.

### 2.2. Ionization States

The earliest direct measurements of ionization states of elements in impulsive and gradual events [62,63] showed $Q_{Fe}$ =14.1 ± 0.2 for gradual SEP events, suggesting a source plasma temperature of ~2 MK, but [3]He-rich events had $Q_{Fe}$ =20.5 ± 1.2 with Si fully ionized. Either the ions in impulsive events came from flare temperatures of >10 MK or the ions traversed enough material after acceleration to be stripped of electrons to an equilibrium charge dependent upon their velocity. The former conclusion conflicted sharply with the abundance enhancements: how can Si/O or Mg/O or Ne/O be enhanced if all these ions have $A/Q$ = 2, like [4]He? This conflict was soon realized [42]. Subsequent measurements by DiFabio et al. [64] showed that the mean ionization state of impulsive Fe did vary with energy, showing the importance of stripping and suggesting that the impulsive events occurred at a depth of ~1.5 $R_S$ where densities were sufficiently high to strip the ions but not greatly disrupt the spectra and abundances of high-$Z$ ions. For comparison, shock acceleration in gradual SEP events begins at 2–3 $R_S$ [65,66].

### 2.3. Theory

The unusual enhancement of [3]He suggests the selective absorption of resonant wave energy, and a large number of possible mechanisms and wave modes have been suggested [67–74] based upon selective heating of [3]He followed by separate acceleration of the enhanced thermal tail by a subsequent unnamed mechanism. However, the mechanism suggested by Temerin and Roth [75] made use of electromagnetic ion cyclotron waves generated by the associated streaming type III electrons to accelerate the ions as they were mirrored in magnetic fields. This mechanism was analogous to that producing the "ion conics" observed in the Earth's magnetosphere.

Unfortunately, however, a resonant mechanism does not produce the continually rising power law of the heavy ions, although it might produce some of the rare abundance anomalies such as extreme enhancement of S seen by Mason et al. [76] in the steep spectra below one MeV amu$^{-1}$. Waves that resonate with [3]He with $A/Q$ = 1.5 could resonate through the second harmonic with S at $A/Q$ = 3, which occurs near 2 MK [10]. Mason et al. [76] saw 16 of these S-rich events in 16 years; also, these abundance anomalies are not seen above ~1 MeV amu$^{-1}$, indicating very steep spectra as discussed in [10].

The enhancement of heavy ions has been explained by Drake et al. [77] as a consequence of magnetic reconnection. These particle-in-cell simulations find the ions to be Fermi-accelerated as they reflect back and forth from mirroring at rapidly converging ends of the collapsing islands of magnetic reconnection, producing strong enhancements vs. $A/Q$. The power of $A/Q$ is related to the power-law width distribution of islands of reconnection. The same physical process is proposed to accelerate electrons in flares [78].

Recently, Laming and Kuroda [79] have suggested that heavy ions in impulsive SEP events could be enhanced as a part of the FIP process. Of course, ions could not also be

accelerated in the dense chromosphere where Coulomb collisions would rapidly remove any energy gained, but this process might enhance heavy ions in a coronal region which would later support jets emitting impulsive SEP events and associated CMEs, both of which would have the strong *A/Q*-dependent enhancements. However, the measured enhancements of impulsive SEPs do not seem to be FIP-biased; Figure 2 shows the average enhancements for 111 impulsive SEP events, measured by Reames, Cliver, and Kahler [46], using different colors to distinguish high- and low-FIP elements.

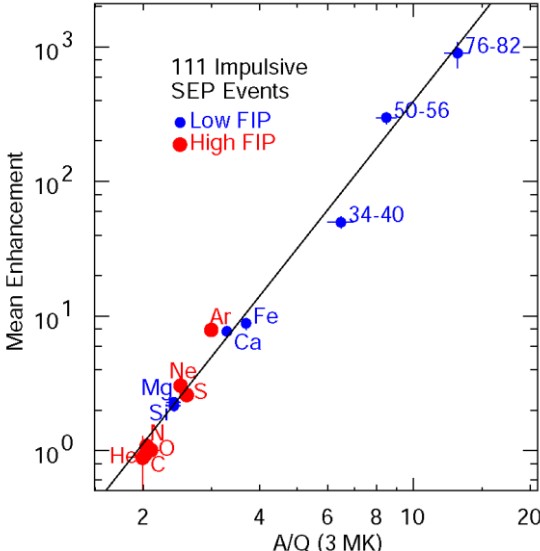

**Figure 2.** Average enhancements (relative to "coronal" abundances derived from gradual SEP events) of element at 2–10 MeV amu$^{-1}$ vs. *A/Q* (with *Q* values at 2.5–3 MK) in 111 impulsive SEP events with elements noted and colors and symbol sizes distinguishing elements with high and low FIP. The average least-squares fit shows a power of 3.64 $\pm$ 0.15 [46] above 2 MeV amu$^{-1}$. Low energy measurements show a power of ~3.26 [45].

In Figure 2, the high-FIP elements Ne, S, and Ar, which began as neutral elements in the photosphere, show a pattern of increase that is no less striking than that of the low-FIP elements Mg, Si, Ca, and Fe that were initially ions. Thus, it seems difficult to conclude that the impulsive SEP enhancements are FIP-related. Furthermore, we are not aware of any similar enhancements being observed in CMEs or other solar-wind plasma that might have sampled these same FIP-enhanced regions. It is much easier to believe that the impulsive-SEP enhancements actually occur in reconnection during acceleration. FIP processes do not drive jets and flares; magnetic reconnection does.

### *2.4. Spatial Transport*

A most distinctive property of impulsive SEP events has been their modest spread in solar longitude in comparison with gradual SEP events. Source longitudes for impulsive events were mainly limited to the W40 to W90 interval, with only a few rare events near E20 [32], while sources of gradual events were spread across the solar disk [19]. This was early evidence for the spatial width of the source shocks in gradual events. These widths depend upon instrument sensitivity for seeing small events, and later observations [46] showed a broader distribution but still few eastern sources for the impulsive events. Sequences of impulsive events often occur as a spacecraft's magnetic connection point scans across an active region.

These source longitude distributions are uncorrected for changes in the Parker spiral with the solar wind speed, a change of 18$^0$ between 400 to 600 km s$^{-1}$. Some of the remaining spread comes from the random walk in the footpoints of the field lines caused by solar surface velocity turbulence discussed by Jokipii and Parker [80]. These field-line distributions exist prior to an event, and they may also include large discrete effects from

the fields carried out by previous CMEs that can produce extensive distortions. Flux tubes, constricted near the Sun, can open out as they expand into the heliosphere. For one event observed by STEREO, Wiedenbeck et al. [81] fit a Gaussian distribution with $\sigma = 48°$.

Harking back to the electrons producing type III bursts and type II bursts in [1], Cliver and Ling [82] actually make use of the differing longitude spans away from their jet and shock electron sources in large events. Shock-accelerated (type II) electrons, poorly-connected, i.e., farther from the source, are correlated with shock-accelerated protons, while well-connected (type III) electrons are not.

Transport along field lines was found early to be nearly scatter-free; Mason et al. [83] fit the angular distributions of several events with scattering mean free paths near 1 AU. Reames, Kallenrode, and Stone [19,84] were able to compare the sharply spiked time duration of a $^3$He-rich event at *Helios* 1 near 0.3 AU with its substantially broadened distribution tracing along the same flux tube to ISEE 3 near 1 AU.

Angular distributions of electrons and ions have been measured since the earliest events [47]. Angular distributions of He, O, and Fe in the 1 May 2000 event considered in Figure 1 are shown in [85].

## 3. Jets and Flares

Early measurements showed no significant CME associations for impulsive SEP events [86], probably because the events are small and the coronagraphs were less sensitive. However, Kahler, Reames, and Sheeley [87] later observed CME associations for larger impulsive SEP events (like the 1 May 2000 event in Figure 1) and related the events to solar jets [88] already associated with type III bursts. Bučík et al. [89–92] soon found many other clear associations of $^3$He-rich events and jets [21].

Solar jets involve magnetic reconnection on open field lines, with a simplified topology shown in Figure 3. Here the rising closed field lines (blue) reconnect with oppositely directed open field lines (black), so the energy in islands of reconnection produces SEPs and CMEs that easily escape to the upper right. The reconnection also forms a newly closed region on the lower left that traps SEPs in a flare where trapped electrons emit X-rays bremsstrahlung. In fact, jets produce associated SEPs, CMEs, and flares, but the SEPs are not accelerated in these flares as was once thought.

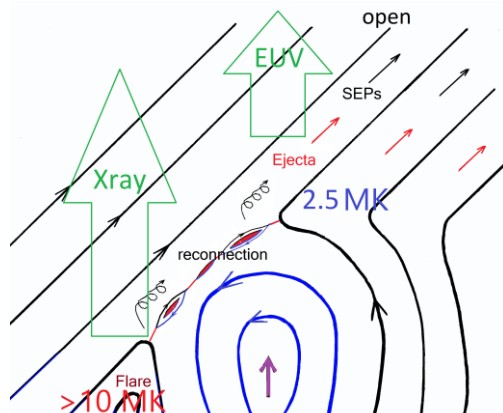

**Figure 3.** Simplified topology of a solar jet shows rising closed field lines of one polarity (**blue**) forming islands of reconnection where they meet oppositely directed open field lines (**black**). SEPs accelerated in the reconnection escape along the open field lines, as does CME plasma. Newly formed closed field regions at the lower left trap energetic electrons and ions and form a flare. This heated (>10 MK) flaring region emits X-rays, while the open 2.5 MK region is observed to be an EUV-emitting region.

Some jets actually have an associated CME with speeds > 500 km s$^{-1}$ that can drive fast shock waves that reaccelerate SEP ions along with local plasma. The speed of the CME in the 1 May 2000 event (Figure 1) is 1360 km s$^{-1}$.

There are much more realistic models of either standard or blowout jets [93–95] that consider CMEs but do not yet consider SEP acceleration. The reconnection and acceleration occurs early, before heating, and, by definition for our purposes, jets always involve open field lines.

Flaring from nearby closed field lines must accompany jets, as shown at the lower left in Figure 3, although the converse is not true since flares may reconnect closed field lines with other closed lines, leaving no path for SEPs to escape. It should be no surprise that flaring on the newly closed field lines in Figure 3 would involve trapped energetic particles with essentially the same abundances as the SEPs that escaped to space—they are accelerated in the same reconnection site. In fact, all flares fed by the products of magnetic reconnection might well involve the same unusual abundances as electron-rich, $^3$He-rich, heavy-element rich, impulsive SEP events. Electron-bremsstrahlung-produced X-rays already suggest electron dominance, as also occurs in the type III radio-emitting impulsive SEP events, but $\gamma$-ray line measurements greatly strengthen the association.

First, Murphy et al. [96] found that broad $\gamma$-ray line measurements suggested that flares could be Fe-rich, like impulsive SEP events. Then in 1999, Mandzhavidze, Ramaty, and Kozlovsky [97] analyzed $\gamma$-ray lines in 20 solar flares, especially the three lines at 0.937, 1.04, and 1.08 MeV from the de-excitation of $^{19}$F* produced with uniquely high cross sections in the reaction $^{16}$O ($^3$He, p) $^{19}$F*, which can be compared with many other lines from excited $^{16}$O, $^{20}$Ne, and $^{56}$Fe, to distinguish $^3$He from $^4$He in the "beam." They found that several of the events had $^3$He/$^4$He~1 and all of them probably had $^3$He/$^4$He > 0.1. More-recently, Murphy, Kozlovsky, and Share [98] identified six key ratios of $\gamma$-ray fluxes dependent upon $^3$He/$^4$He in the beam; all these ratios showed increased $^3$He with an average $^3$He/$^4$He ratio of 0.05–3.0. These studies involve ~135 de-excitation lines from products in ~300 proton and He-ion reactions. $^3$He-rich events produce a distinctly different pattern of $\gamma$-ray lines [98]. These $\gamma$-ray lines were all measured in large flares, not in small jet-associated events, suggesting that flares typically have abundances that we associate with impulsive SEP events from jets. Yet, SEPs from these flares (e.g., products of the reactions like $^2$H, Li, Be, and B) are not actually seen since they are all trapped on closed loops where they interact to produce the $\gamma$-rays that are seen.

## 4. Abundance Power Laws, Temperature, and Shocks

To obtain the power law seen in Figure 2, we have normalized average impulsive-SEP abundances to "coronal" abundances from gradual SEP events and have used a source temperature that produces reasonable ionization states $Q$ for the elements. It was realized much earlier [42] that only the temperature range of 3–5 MK would produce similar $A/Q$ values corresponding to the similar enhancements observed for Ne, Mg, and Si [42]. Suppose we now assume that the observed element enhancements *must* form a power law in each individual SEP event. We can then find a temperature that produces the best fit, i.e., try all temperatures in a wide range and pick the best fit [34,99–102]. Such an analysis for an observed event is shown in Figure 4. Figure 4b shows fits of the same measured enhancements (each shifted by a decade) plotted vs. $A/Q$ using $Q$ values for the five temperatures listed. The $\chi^2/m$ values of these least-squares fits are plotted vs. temperature in Figure 4c, selecting 2.5 MK as the "best" by a small margin; 3.2 MK is close. The original application of this technique to the 111 impulsive SEP events [99] found 79 events at 2.5 MK and 29 at 3.2 MK, i.e., not much variation. Subsequently, Bučík et al. [91] found extreme ultraviolet (EUV) temperatures in jets leading to $^3$He-rich events to be 2.0–2.5 MK, in reasonable agreement with those found from the best-fit technique using SEP element abundances.

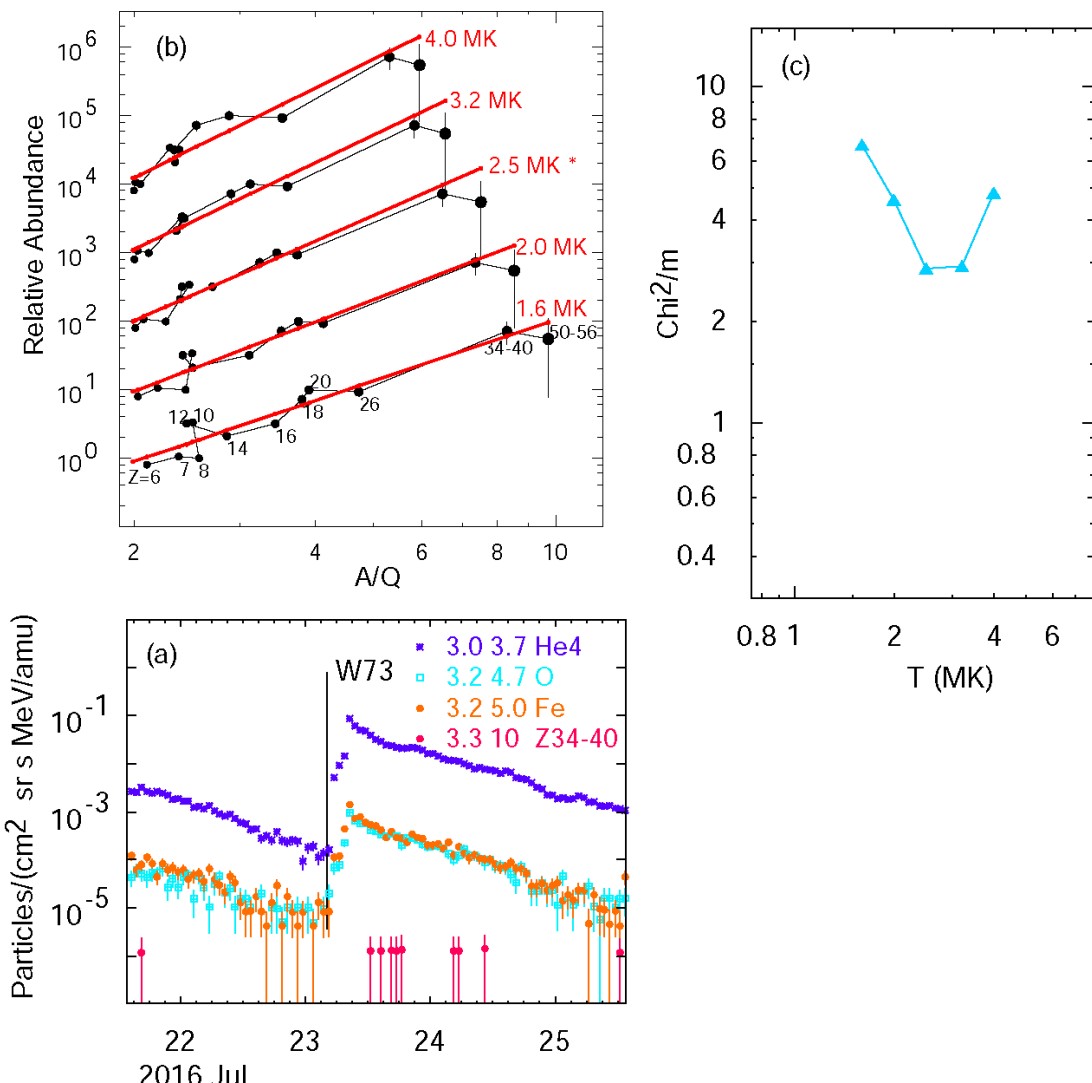

**Figure 4.** Panel (**a**) shows time histories of $^4$He, O, Fe, and $34 \leq Z \leq 40$ at the indicated MeV amu$^{-1}$, for an impulsive SEP event beginning on 23 July 2016. (**b**) Shows power-law fits to the abundance enhancements of elements with $Z \geq 6$ at five different temperatures, *T*, with *Z* values of the elements noted at the lowest *T* (1.6 MK). Only the *Q*-values change with differing *T*. (**c**) Shows $\chi^2/m$ values for the five fits plotted vs. temperature.

Figure 5 shows the analysis of a series of three impulsive SEP events. Event numbers refer to the event list in [46]. Proton abundances have been included in these plots but not in the fit. For the first two events, Figure 5c shows that the extension of the power-law fits for the elements with $Z \geq 6$ passes very close to the measured proton and $^4$He abundances. However, for Event 5 there is suddenly a significant proton excess, labeled in Figure 5c and clearly noticeable around the arrow in Figure 5a. Reames [35] identified four SEP abundance patterns, SEP1 events are "pure" impulsive events with power laws extending to protons, while SEP2 events have a significant proton excess like Event 5. Patterns SEP3 and SEP 4 refer to gradual events with (SEP3) and without (SEP4) reaccelerated impulsive seed particles dominating their high-Z regions.

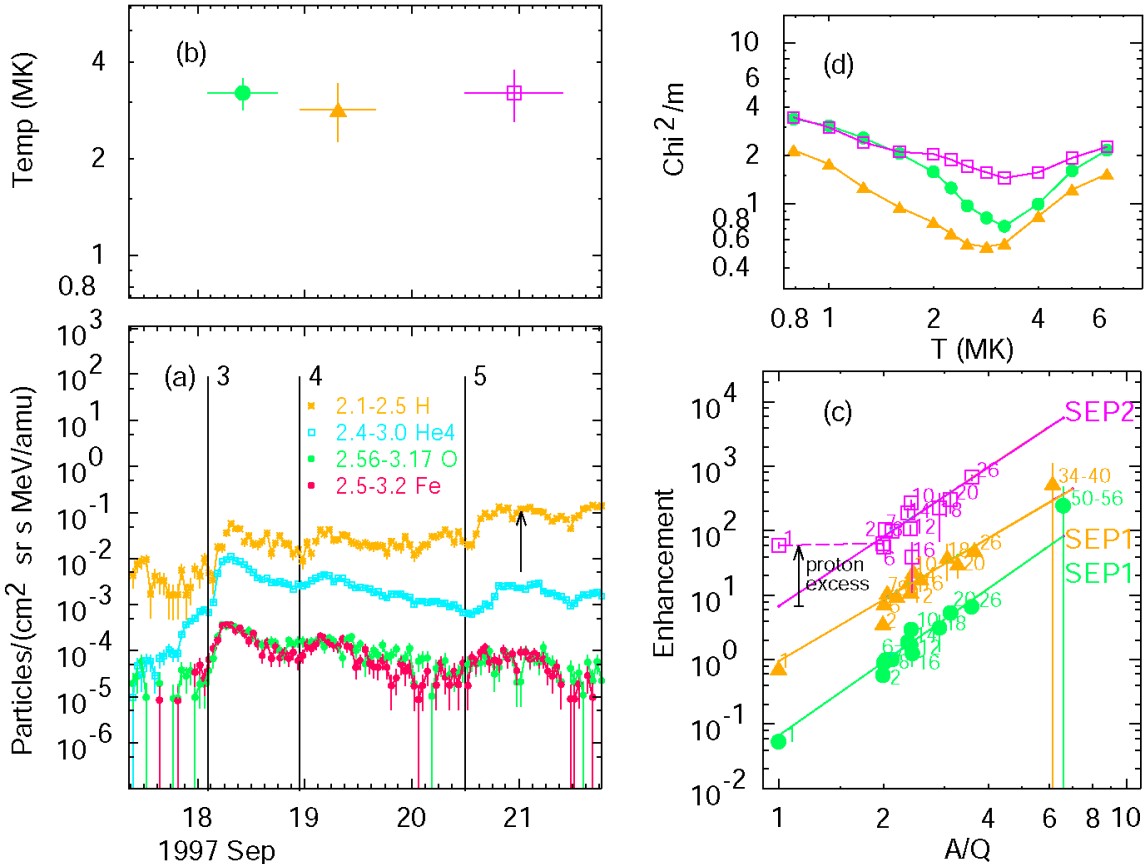

**Figure 5.** Panel (**a**) shows intensities of H, $^4$He, O and Fe at the listed energy in MeV amu$^{-1}$ for a sequence of three small impulsive SEP events numbered as in the event list of [46] so earlier work on these events may be accessed, (**b**) shows the derived best-fit temperatures for each, (**c**) shows the corresponding best-fit power-law abundance enhancements, with the measurements for each element labeled by $Z$, and (**d**) shows $\chi^2/m$ vs. temperature for each event. The three events are distinguished by symbol and color in panels (**b**–**d**). Only elements with $Z \geq 6$ are included in the fits.

Unfortunately, no CME is observed for Event 5, but a more extreme example of a proton excess in a SEP2 (or SEP3) event is Event 92, shown in Figure 6. In this event, which has a 925 km s$^{-1}$ associated CME, we assume the shock samples the pre-enhanced impulsive SEPs, causing them to dominate high $Z$, while ambient H and $^4$He dominate at low $Z$, as an explanation of the behavior that cannot be explained by a single line of enhancement. Other ambient ions such as C and O would have higher $A/Q$ at lower ~1 MK coronal temperatures and would contribute little because of the declining slope. The suppressed impulsive-SEP H also does not contribute. The observed $^4$He may have a contribution from both seed populations in this event. This is not merely an enhancement that begins for elements heavier than $^4$He where H and $^4$He are at the same level as Event 5 in Figure 5c; here, for Event 92 in Figure 6c, protons have an order-of-magnitude greater enhancement than $^4$He, C, or O. The event is most likely SEP3 because it has a strong shock and is preceded by an event at the same location with similar heavy-element enhancements, i.e., SEP1 enhancements can come from a pool that precedes the event; SEP2 enhancements come from the same event.

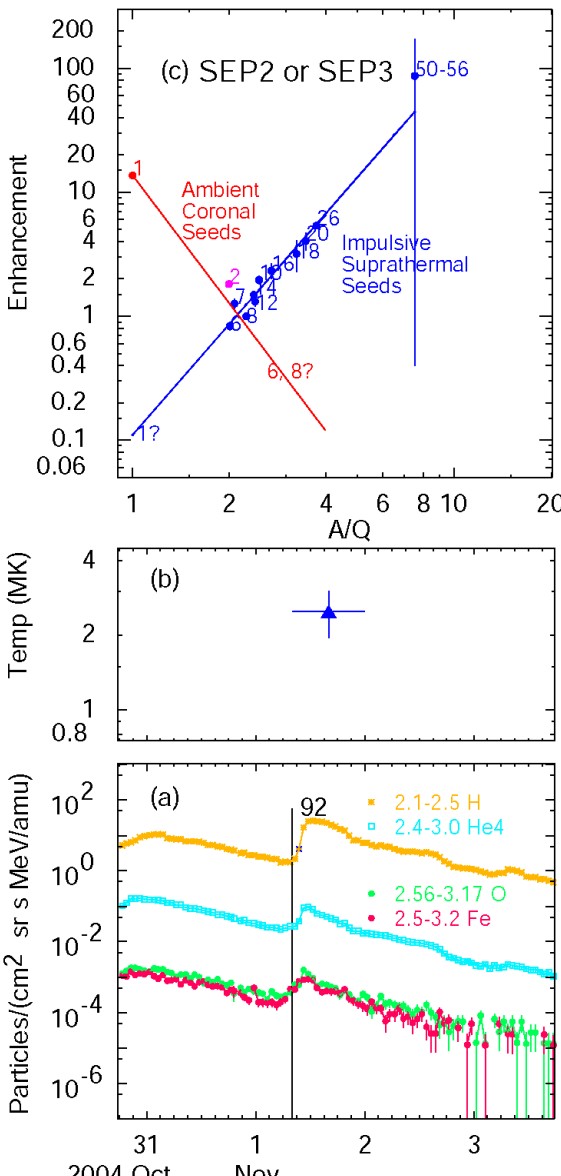

**Figure 6.** Panel (**a**) shows intensities of H, $^4$He, O and Fe as in Figure 5 for impulsive SEP event number 92 in the event list of [46], (**b**) shows the derived temperature, (**c**) shows the corresponding best-fit power-law abundance enhancements in **blue** with the measurements for each element labeled by Z, only elements with Z $\geq$ 6 are included in the fit. The labeling postulates shock reacceleration of impulsive SEP1 seed particles in **blue** and mostly ambient coronal seed particles (H and $^4$He) in **red**. This event has an associated 925 km s$^{-1}$ CME.

The theory of Drake et al. [77] allows for the enhancements to begin at a higher value of *A/Q* than 1, so Event 5 in Figure 5 need not involve shock acceleration. However, there is a strong tendency for larger impulsive SEP events to have fast CMEs as in shown in Figure 7. Figure 7 shows peak proton intensity vs. proton excess, with CME speeds as the symbol, when available. Events with large proton excesses tend to be larger events that have fast CMEs. However, some events with large proton excesses have no associated CME observed (small blue circles). Could this be an observational sensitivity problem?

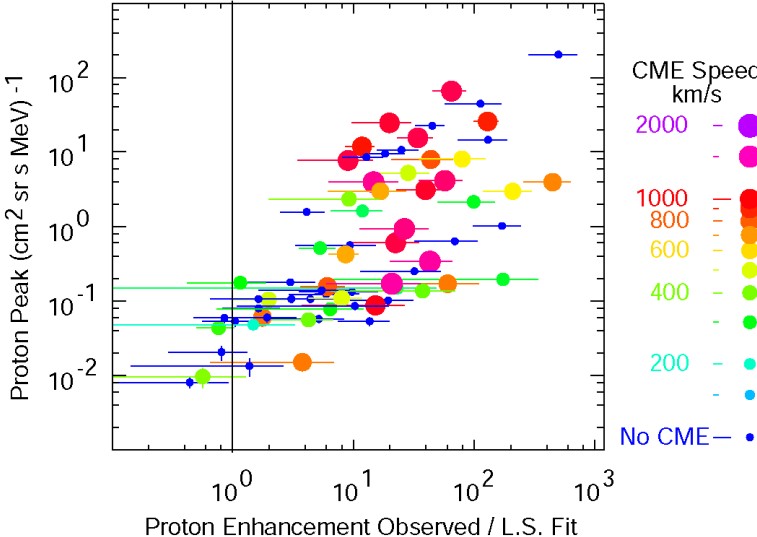

**Figure 7.** Shows the correlation of impulsive SEP peak proton intensity vs. proton excess with CME speed denoting the symbol size and color. Events with fast CMEs all have large proton excesses; so do some events with no visible CME.

Thus, we see three possible power-law patterns for abundance enhancements in impulsive SEP events:

(1)  A single power-law fit extends from H to high-Z elements (Events 3 and 4 in Figure 5). This is seen for many small impulsive SEP1 events.

(2)  A large proton excess, sometimes including an enhancement in $^4$He, with an associated fast CME, that shows a clear SEP2 event, with a fast CME-driven shock (Event 92 in Figure 6) that accelerates ions from both seed populations.

(3)  A proton excess, roughly equal with the level of $^4$He, and no fast CME (Event 5 in Figure 5); this could be either a SEP1 event where the enhancement happens to begin above C and O (possible in Drake et al. [77]) or a SEP2 event where the coronagraph failed to show the fast CME. A puzzle.

Event 92 in Figure 6 cannot be explained by shifting the onset of the high-$A/Q$ enhancement, as could Event 5, where H and He have equal enhancements of ~1. Once we believe the enhancements of all elements from a source must fit a specific power-law pattern, then protons that have strong alternate enhancements must not come from that same source.

The abundance of He is complicated since it can be included with the heavy ions in a SEP1 event, or with the protons in a SEP2 event; i.e., He could be dominated by the impulsive component or by the ambient H seed component. Worse, we also find that about ~10% of the impulsive SEP1 events have $^4$He depressed by a factor of ~10 [103–105], perhaps because it has the highest FIP (24.6 eV) among the elements and failed to be adequately transported [106] into the local coronal underlying these particular SEP events. Having little to add that is new, we leave the discussion of $^4$He abundance to a previous comparative consideration of both impulsive and gradual SEP events [105].

The assumption that Z $\geq$ 6 abundances vary as powers of $A/Q$ has worked very well at energies above ~1 MeV amu$^{-1}$, although there have been a few exceptions that have contributed to small events at lower energies [10,76]. There have been studies to see if apparent variations in abundances such as Ne/O could be significant [104] or could result from spectral differences, for example, but no systematic variations could be found and statistical fluctuations could not be excluded. The energy spectra in these events are also approximately power laws in energy per nucleon above ~1 MeV amu$^{-1}$ [104]. The resonant processes that enhance $^3$He could certainly have interesting low-energy consequences at higher Z, but these processes have had no apparent effect on any of the events above ~2 MeV amu$^{-1}$ that have been observed by the *Wind* spacecraft with nearly continuous

coverage since 3 November 1994. These power-law fits at high $Z$ have provided a firm basis that allows us to contrast and highlight the separate behavior of H and He.

We have considered abundances in impulsive SEP events relative to coronal (average gradual event) abundances to see the power-law dependence. If we consider abundances in a single impulsive event relative to the impulsive event average, the large variations in H and $^4$He stand out more, since Fe/O is a nearly constant signature of impulsive SEP events [46]. Since most impulsive events, especially the large ones, are SEP2 events (requirement of measurable abundances of Fe and rarer elements tends to select larger events), the average favors SEP2 events so that SEP1 events become characterized by large proton suppressions. Fitting as a power law is no longer relevant when the reference is SEP2 events rather than the corona. Suppressions of $^4$He in He-poor events [103–105] also stand out in such a comparison.

## 5. Conclusions

Did the study of impulsive SEP events begin with the observation of type III radio bursts in the 1960s, with the observation of $^3$He-rich events in the 1970s, or with the joining of the two in 1985? Impulsive events began to have their own identity, beyond being the "first phase" of large flares [107]. Finding the large resonant-wave enhancements of $^3$He was a surprise. Finding the contrasting smooth rise in heavy elements up to Fe was another advance, made more conclusively a power law in $A/Q$ with the observations of heavier elements up to Pb (Figure 2). The association of impulsive SEP events with solar jets provided concrete sources for the events and implicated magnetic reconnection on open field lines in the physics of enhancement and acceleration. The associated CMEs, which sometimes drive fast shocks, add interesting new complexity to the acceleration physics, complexity we can address by serious consideration of the consequences of large "proton-excess" abundances. The evidence of an alternate source of protons in these proton excesses suggests that energy spectra from the alternate source might be seen in low-energy spectra.

It is important to recognize that flares and impulsive SEP events share similar physics of acceleration, allowing us to explore details of the physics of the element abundances and spectra produced in SEPs that are completely inaccessible with X-ray and $\gamma$-ray observations of flares alone. X-rays tell us nothing about energetic ions. SEP abundances provide an insight into the physics of magnetic reconnection, which is also a fundamental process in the physics of solar flares. Source plasma temperatures derived from the SEP abundances show no evidence of heating before or during acceleration—a new revelation when applied to flares. Thus, reconnection must occur early, rapidly, and at low density to avoid energetic ion heating. A combination of SEP and $\gamma$-ray-line results might improve our understanding of both. We can also learn by comparison with the shock-accelerated gradual SEP events that can wrap around the Sun; they have benefited greatly from multispacecraft studies [19].

It is important to emphasize that the SEP source plasma temperatures are ~2.5 MK. There is *no* evidence of heating of the source plasma during reconnection and acceleration of SEPs. As applied to flares this means that reconnection and particle acceleration occur rapidly, early, and at low density, before much heating. Being magnetically trapped, flare particles then go on to scatter into the loss cone, stopping and heating the chromosphere, which expands, filling loops with bright, hot chromospheric material that emits soft X-rays.

A great hope for the new spacecraft that approach the Sun more closely, Parker Solar Probe and Solar Orbiter, is that they can provide time profiles of impulsive SEP events with time resolution more comparable with those of solar X-rays or type III radio bursts. SEP time profiles at 1 AU are blurred and extended by scattering and delays in transit. We could begin to identify, with greater certainty, which of the many small type III burst or X-ray peaks in Figure 1 goes with which $^3$He-rich event and ask why some events contribute electrons and $^3$He while others, from the same active region, apparently do not. There are already interesting studies, mainly of the electron, i.e., type III, timing and spatial distributions [108,109].

Another potential study of events closer to the Sun involves low-energy ($<1$ MeV amu$^{-1}$) spectra that may flatten or roll over if particles have traversed 50–100 μgm cm$^{-2}$ of coronal plasma from deeper sources. Matter traversal that has stripped electrons from the Fe [64] may or may not be enough to modify the spectra differently for H, $^4$He, O, and Fe. Do the depths of events differ? For events with fast shocks, is there any evidence of reacceleration or an additional un-attenuated H spectrum from shock acceleration?

Energetic-particle abundance data presented here are from the Low Energy Matrix Telescope (LEMT) on the *Wind* spacecraft [110]. These data are available at the NASA Coordinated Data and Analysis Web site: https://cdaweb.gsfc.nasa.gov/sp_phys/ (accessed date 5 October 2023).

**Funding:** No institutional funding was provided for this work.

**Acknowledgments:** The author thanks Radoslav Bučík for a useful discussion of proton excesses.

**Conflicts of Interest:** The author declares that this research was conducted in the absence of any commercial or financial relationships that could be construed as a potential conflict of interest.

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
