# Peer review of "Element Abundances in Impulsive Solar Energetic-Particle Events"

_universe, doi:10.3390/universe9110466_

Round 1
Reviewer 1 Report
Comments and Suggestions for Authors
This paper is a very good review on abundances in impulsive solar energetic particles. It presents both an historical overview as well as a good description of the present knowledge of the topic.
I would however suggest that the author discusses even briefly the potential advances that could be brought to this topic by the new solar probes in the inner heliosphere.
A few minor corrections:
There are 2 subsections 2.3.
Line 213: reference 92 is not Bucik et al
Figure 5 : The dates of the events do not show well
Author Response
See file

Reviewer 2 Report
Comments and Suggestions for Authors
I have thoroughly reviewed the paper titled "Element Abundances in Impulsive Solar Energetic-Particle Events" authored by Donald V. Reames. The paper is well-written, and its quality is good. However, incorporating the following suggestions would further enhance its overall quality:
The paper investigates impulsive solar energetic-particle (SEP) events, which are distinct from shock-induced type II bursts and are characterized by their unexpectedly high 3He content. The paper reveals that these events exhibit element abundance patterns that smoothly change with the mass-to-charge ratio A/Q, extending up to elements like lead (Pb). These SEPs originate during magnetic reconnection processes in solar jets where open magnetic field lines enable the escape of energetic particles. This differs from impulsive solar flares where similar reconnection occurs within closed field lines, trapping energetic ions that release their energy as X-rays, γ-rays, and heat. The paper introduces the concept of power-law relationships in abundance enhancements relative to A/Q, offering insights into the temperature of the coronal source in these events. Additionally, the presence of proton and helium excesses suggests the possibility of re-acceleration of SEPs by shock waves generated during accompanying narrow coronal mass ejections (CMEs) in certain solar jets.
Elaborate on how type III radio bursts and 3He-rich SEP events are linked and how they relate to the broader understanding of impulsive SEP events.
Provide more details on the characteristics and properties of 3He-rich events in terms of their element abundances and energy spectra.
How are the element abundances observed in 3He-rich events related to the A/Q ratio, and how does this trend compare to previous models?
Expand on the implications of proton excesses and how they suggest the presence of an alternate source for protons.
Provide more specific details on how the physics of element abundances and spectra in SEPs differ from X-ray and γ-ray observations of solar flares.
What is the significance of velocity dispersion in electrons and ions, and how does it relate to the behavior of 3He-rich events?
Provide insights into the cases where electron trapping is minimal, as mentioned in the text.
Upon careful examination, I have observed that the manuscript contains more than 40 instances of self-citation. While self-citation is acceptable when it is necessary to provide context or reference prior related work, such a high number of self-citations may raise concerns about potential bias and the objectivity of the research presented. I recommend that the authors reconsider their self-citation practices and strive for a more balanced and diverse set of references. This will not only enhance the credibility of their work but also contribute to a more comprehensive and well-rounded discussion of the topic.
Comments on the Quality of English Language
The English quality of the paper is generally good, with clear and concise writing that effectively conveys the research findings. However, there are a few minor grammatical and language issues that should be addressed to ensure the manuscript maintains a high standard of clarity and precision.
Author Response
See file

Reviewer 3 Report
Comments and Suggestions for Authors
The paper presents an exhaustive discussion on impulsive Solar Energetic-Particle Events, based on a wide bibliography now existing on the matter. This is considered a valuable paper and is recommended for publication. Minor revisions are encouraged.
In what follows there are a few comments that the author is encouraged to consider and take into account.
I do not completely share the interpretation of the jets as phenomena separated by the flare itself. During the magnetic reconnection process, that can take place according to several different 3D configurations of the reconnecting fields, the flare starts with energy release, due the reconnection process, accompanied by plasma jets, very localized heating, non-thermal electrons/hard X-rays; as a consequence, the chromosphere heats up, by thermal conduction or impinging electrons, and fills the closed loops with hot chromospheric material, forming the emitting soft X-ray source. Indeed, the peak in hard X-rays precedes the soft X-ray one, but they are part of the same flare process. Not necessarily jets are observed/observable. (See, for instance, how reconnection during flares was discussed since early publications such as the monograph ‘Physical processes in solar flares’, B. V. Somov, 1992, Kluwer.)
Thus, some statements in Sect. 3 on jets vs. flares should be rephrased in my opinion, being not necessarily separated phenomena but different aspects of the same process. In addition, the flare plasma observed in soft X-rays is fed in closed loops only indirectly by magnetic reconnection, since this is the process that releases the flare energy, inducing chromospheric heating at the loop footpoints and the subsequent chromospheric evaporation.
In addition, it would be helpful for the reader to broaden somewhat the Conclusions section reporting a bit more extensively the various points.

Author Response
See file

Round 2
Reviewer 2 Report
Comments and Suggestions for Authors
Dear Donald V. Reames
Thank you for your detailed response and for sharing your perspective on the matter of self-citation. I appreciate the depth of your experience in the field, which spans nearly sixty years and includes an extensive body of work.
Your points about the role of self-citation in providing depth and context to the research are well-taken. I understand that your extensive contributions to the field naturally lead to situations where citing your previous work is both relevant and informative.
Given your explanation, I consider the level of self-citation in your manuscript to be justified. I appreciate your willingness to engage in this dialogue, and I look forward to seeing your research continue to shape the field.
Author Response
Thanks for your comments.